# Adaptive and Scalable Discovery and Mitigation of Multiple Biased Subgroups in Image Classifiers

## Abstract

Deep learning models generally perform well across entire datasets but often exhibit disparate behaviors across different subgroups. Such biases hinder real-world applications. Despite numerous efforts to identify and mitigate biases in biased subgroups using the powerful vision-language foundation model CLIP, these approaches commonly neglect inherent biases in CLIP's feature encoding, which can restrict performance improvements. In our work, we introduce a novel strategy that employs an ensemble of surrogate models for adaptive and scalable discovery of biased subgroups, effectively reducing the impact of feature encoding biases inherent in CLIP. Additionally, we utilize the large vision-language model to elucidate inherent subgroup biases and employ relative Fisher information to identify critical layers for mitigating subgroup bias and suppressing the learning of shortcuts. Extensive experiments on CIFAR-100, Breeds, and ICSD-171K demonstrate the effectiveness of our proposed methods. We also confirm the presence of subgroup bias by analyzing the image encoder of CLIP on the Hard ImageNet dataset.

## 1  Introduction

While deep learning models have presented significant progress in many areas, such as image classification (Rao et al., 2021; Chen et al., 2021), text-to-image generation (Zhou et al., 2022; Li et al., 2023), and robotics (Károly et al., 2020; Chen et al., 2020), there are also some concerns regarding the reliability of their use. These concerns include adversarial vulnerability (Zhang et al., 2024b; Wang et al., 2023), privacy leakage (Tyagi & Goyal, 2020; Yan et al., 2021), and subgroup bias (Zhang et al., 2024a; Jain et al., 2022). Among these issues, subgroup bias is one of the most prevalent trustworthiness problems in model development. Taking the image classification task as an example, the well-trained model usually achieves good overall performance on the known class divisions but performs poorly on the unannotated subgroups within each class (Zhang et al., 2024a). Discovering and mitigating these biased subgroups is crucial for understanding model failures and improving trustworthiness (Zhang et al., 2024a; Jain et al., 2022; Eyuboglu et al., 2022).

In recent years, many research works have been put into efforts on bias discovery but with limitations. Most of the existing works focus on discovering the bias from known attributes, leading to unknown bias overlooked. Some research focuses on unknown bias discovery. Li et al. (2022) designs a debiasing alternate network, which predicts the subgroup label, but can only discover a single bias and needs human explanations for the subgroup. For automatic bias discovery and explanation, current works usually employ the cross-modal capacity of the vision-language foundation model (CLIP) (Radford et al., 2021) for bias discovery and interpretation. Eyuboglu et al. (2022) leverages the CLIP embedding to represent the samples and proposes an error-aware mixture model for clustering to group the under-performing slices and identify the systematic error, but the use of proxies of clusters can result in interpretation distortion. Jain et al. (2022) further propose to use a linear classifier to decompose the CLIP space for single biased subgroup explanation, while Zhang et al. (2024a) further proposes the partial least square method for CLIP space decomposition to multiple biased subgroup discovery and integration. However, all of these works assume the CLIP learns unbiased latent space by training on large-scale datasets. This raises a natural question (**Q1**) to us: *Is CLIP itself reliable enough to encode the features without bias?*

We conduct an experiment here to answer this question. Specifically, we fine-tune a CLIP followed by a linear layer for the image classification task on the whole ImageNet dataset (♠). To avoid introducing novel bias during fine-tuning, we freeze the parameters of CLIP and only optimize the linear layer. To better understand the influence of bias, we used entropy to indicate the confidence of the CLIP decision, where the drop in confidence reflects the impact of the spurious correlation on this decision. By evaluating the model performance on Hard ImageNet, which is the subset of ImageNet consisting of more spurious correlation, it has a minor drop performance of 5.4% (59.7% *v.s.* 54.3%) with nearly all classified correctly by masking the target objects (second column in fig. 1). However, on the target object (third column in fig. 1), it only has a classification accuracy of 0.7%. Besides, even with correct classification, there is still a large confidence degradation with up to 48.9% (see the example of "dog sled"). These results suggest that CLIP is also biased by spurious correlation existing in the datasets, which potentially influences the reliability and robustness of previous biased subgroup discovery work based on CLIP.

Furthermore, the task of discovering multiple biased subgroups faces two significant challenges. First, the number of subgroups within each class is unknown. Previous studies have assumed a predefined, equal number of subgroups in each class for biased subgroup discovery (Jain et al., 2022; Zhang et al., 2024a), an approach that lacks adaptivity for real-world applications. Second, after identifying biased subgroups, it is crucial to understand the distribution shifts between them. This understanding helps to better explain the inherent biases and to mitigate them, thereby improving model robustness.

Figure 1: CLIP is not perfect, which also learns the bias, *e.g.*, the spurious correlation. We use "Cf." to denote the decision confidence and "Cf. Drop" to denote the confidence drop. The ✓ and x indicate whether the image is classified correctly.

In our work, we empirically find that even the studied biased model can be directly deployed to accurately find multiple biased subgroups without the help of CLIP. Based on this observation and the fact that different model has similar but different interests of the region for one image, we propose to use diverse surrogate models to replace CLIP in adaptively and scalably discovering and mitigating the multiple biased subgroups. Specifically, we first pre-train different small surrogate models on different datasets and then fine-tune them on the studied dataset. By such design, models can recognize images from different views, enabling the cross-model compensation of their biased feature encoding. By adaptively decomposing the latent representation of the feature with model supervision, we derive the eigenvectors of different subgroups, which are used to generate pseudo subgroup labels for images. Then, we propose to leverage the large vision-language model to help us understand the distribution shifts between different subgroups as well as the existence of multiple biased subgroups. We define a metric of relative layer importance score to find the layer for subgroup biased mitigation efficiently and surpress the overlearning of subgroups with shortcuts.

Our contributions are summarized as follows:

1. We propose the use of an ensemble of diverse models to replace CLIP for the adaptive discovery of multiple biased subgroups, enhancing the adaptivity and precision of bias detection.

2. We leverage a large vision-language model to facilitate a deeper understanding of the distribution shifts between different subgroups, aiding in the explanation and mitigation of inherent biases.

3. We introduce a novel metric to identify critical layers responsible for learning biased subgroups and suppressing shortcut learning. We then fine-tune these layers to efficiently mitigate bias.

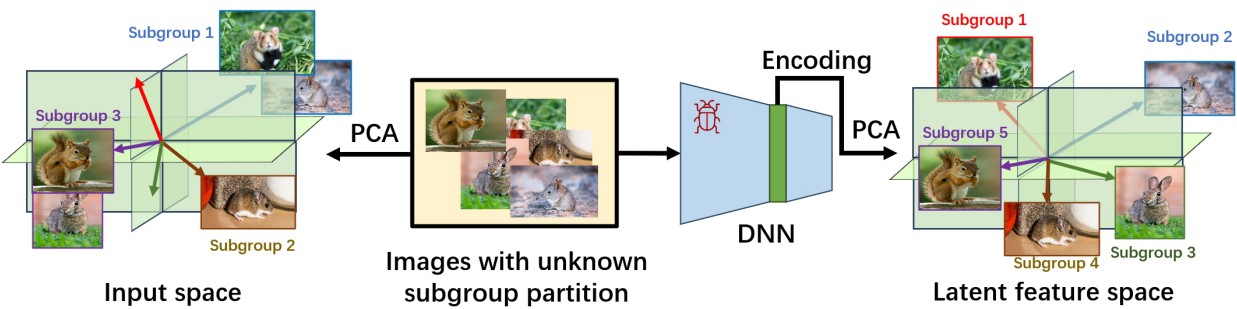

Figure 2: Decompose the inputs or the encoded features to discover the multiple subgroups.

4. We conduct extensive experiments on the CIFAR-100, Breeds, and ICSD-171K datasets, which demonstrate the effectiveness of our approach. Additionally, we confirm the presence of subgroup bias in the CLIP feature encoder by analyzing the Hard ImageNet dataset.

## 2    Related work

**Bias Identification**    Numerous studies have focused on discovering and interpreting biases in machine learning models. Eyuboglu et al. (2022) employ the cross-modal embeddings of the CLIP model to pinpoint underperforming sample subsets. Singla & Feizi (2022) find that the activation maps of neural features can be used to spotlight spurious or core visual features. Zhu et al. (2022) leverage a training-free framework aimed at elucidating dataset-level distribution shifts. Generative models have been employed by Li & Xu (2021); Lang et al. (2021) to unearth and interpret unknown biases. Jain et al. (2022) utilize a hyper-plane to identify model failure modes, utilizing CLIP for automatic captioning to elucidate these modes. Zhang et al. (2024a) discover multiple biased subgroups using supervised principle component analysis in latent representations.

**Bias Mitigation**    Various approaches have been proposed to mitigate bias in machine learning models. These encompass re-sampling and weighting strategies Li & Vasconcelos (2019); Qraitem et al. (2023), distributional robust optimization techniques Słowik & Bottou (2022); Wen et al. (2022), invariant risk minimization Mao et al. (2023), and adversarial debiasing methods Zhang et al. (2018); Lim et al. (2023). Some methods utilize identified biases to counteract model bias, such as EIIL Creager et al. (2021) and LfF Nam et al. (2020a). However, these approaches often necessitate bias labels in the training dataset, which are typically absent in practice, thus limiting their scalability. To address this challenge, recent research has explored bias mitigation without access to bias annotations. Nam et al. (2020b) propose training a debiased model against a prejudiced model. Pezeshki et al. (2021) introduce a regularization term to decouple failure learning dynamics. Li et al. (2022) present debiasing alternate networks to uncover and unlearn multiple identified biases in classifiers. Park et al. (2023) propose debiased contrastive weight pruning to develop unbiased networks. Zhang et al. (2024a) propose the soft-label strategy for boosting the conventional bias mitigation methods.

## 3    Methodology

### 3.1    Motivation

The key challenge in discovering multiple subgroups lies in identifying subgroup-specific discriminative features that the model inadvertently learns—features that remain consistent within a subgroup but differ across subgroups. Previous work (Eyuboglu et al., 2021; Jain et al., 2022; Zhang et al., 2024a) has demonstrated the use of foundation models, such as CLIP, as encoders to obtain latent representations for subgroup discovery. However, this approach introduces two additional challenges: (1) foundation models themselves carry implicit

biases, which can hinder the effectiveness of subgroup discovery, and (2) relying on large models to debias smaller models limits scalability when attempting to debias the large models themselves.

To find solutions to these challenges, there naturally raises a question, *can we only use the studied biased model without any supplementary to discover multiple subgroups?*

Previous works have identified that it is impossible to derive invariant features from the heterogeneous data without environmental information, which motivates their design of CLIP feature decomposition with the model supervision. Leaving the black-box setting alone, we directly use the feature computed by the studied biased model for the decomposition step to discover subgroups, which also serve as the model supervision. Specifically, in example (♣), we forward all images of "small mammal" to ResNet-101, get the features computed by the last max pooling layer, and merge these features together to a matrix followed by the decomposition step using the PCA with setting 5 as the number of principal components. We use the obtained 5 principle components to compute, respectively, the similarity of each image to each subgroup indicator, *i.e.*, the principle components, and get the pseudo subgroup label that achieves the maximum similarity, for each image. We also set up an additional experiment by directly decomposing the group of inputs to discover multiple subgroups. For evaluation, we report the ratio of the most frequent pseudo-subgroup label in each ground truth subgroup of the superclass "small mammals".

The whole pipeline of the aforementioned experiment and results are shown in fig. 2. It can be seen that even setting 5 principle components, the direct decomposition of inputs by PCA can only find 3 subgroups, in which the "mouse" and "hamster", "rabbit" and "squirrel" are respectively wrongly mixed into the same subgroup. By comparison, the decomposition of computed features after the max pooling layer of ResNet-101 can be accurately used to discover five different subgroups. It suggests that even without explicit information about subgroups during the training process, the feature representation learned by the model is implicitly influenced by the distribution shifts between different subgroups, which can be directly employed for subgroup discovery without any supplementary.

Motivated by these findings, we propose to model subgroup information using an ensemble of diverse small models and derive invariant features to discover and mitigate multiple biased subgroups.

### 3.2 Adaptive Biased Subgroup Discovery

We begin by pre-training $N$ distinct models, denoted as $\{f_n\}_{n=1}^N$, on various datasets before fine-tuning them on the target dataset $\mathcal{D}$. Each model $f_n$ is composed of two components: $f_n = g_n \circ \phi_n$, where $g_n$ represents the feature encoder, and $\phi_n$ is the classification head. This architecture encourages the model to interpret the data in $\mathcal{D}$ from diverse perspectives. For instance, a model pre-trained on the Describable Textures Dataset (Cimpoi et al., 2014) and then fine-tuned on CIFAR-100 can utilize prior texture knowledge to classify novel images more effectively.

Next, we pass data from a specific class through the $N$ different encoders $\{g_n\}_{n=1}^N$, generating encoded features $\{g_n(X)\}_{n=1}^N$, where $g_n(X) \in \mathbb{R}^{S \times d_n}$. Here, $S$ denotes the number of samples in $X$, and $d_n$ is the dimensionality of the encoded feature for the $n$-th model. Simultaneously, we forward the data through the black-box model under study, obtaining a supervision signal $P \in \mathbb{R}^{S \times M}$, where $M$ represents the number of supervision signals (e.g., logits, loss values, or other forms of supervision).

To dynamically identify distinct subgroups within this class, we project the concatenated encoded features $G_X = \{g_n(X)\}_{n=1}^N \in \mathbb{R}^{S \times \sum_{n=1}^N d_n}$ onto the supervision space defined by the black-box model, represented by $P$. This projection is achieved through a dynamic singular value decomposition (SVD) process that determines the optimal number of subgroup bases based on a reconstruction error threshold. Specifically, we aim to decompose $G_X$ into a basis matrix $U \in \mathbb{R}^{D \times g}$ and a coordinate matrix $\Sigma \in \mathbb{R}^{g \times M}$ such that each column of $U$ represents a subgroup basis vector, and $\Sigma$ contains the coordinates of these basis vectors in the supervision space.

The optimal rank $g$, corresponding to the number of subgroups, is determined dynamically by controlling the reconstruction error of the low-rank approximation. We compute the SVD of $G_X$ as $G_X \approx U \Sigma V^T$ and

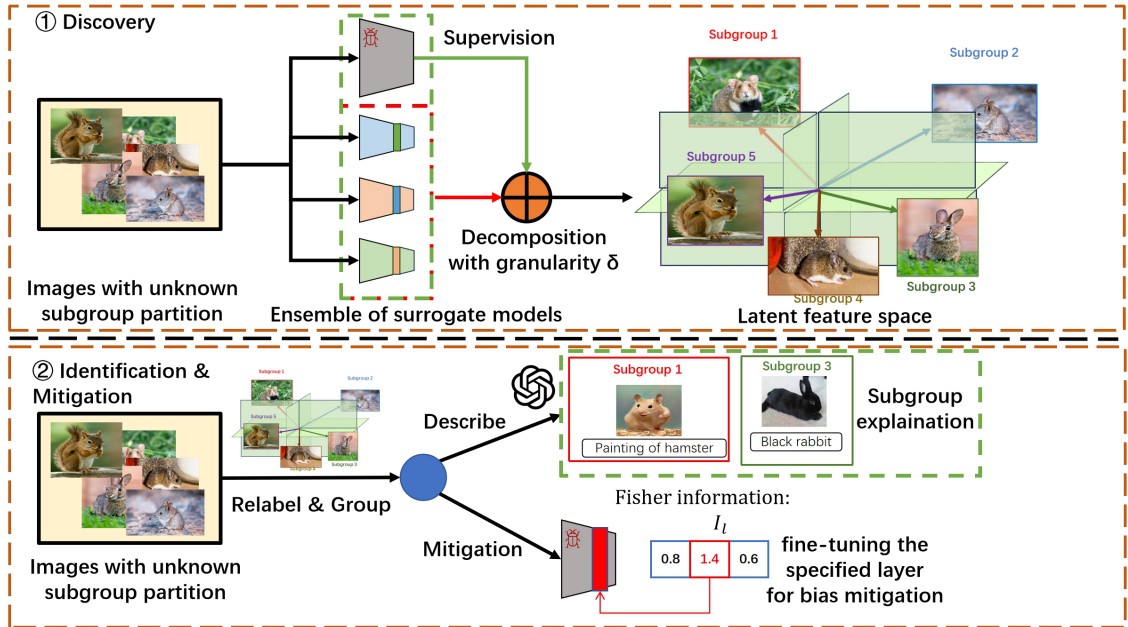

Figure 3: Overview of our proposed method: It consists of three components: (1) adaptively discovering multiple subgroups with decomposition tolerance $\delta$, (2) explaining the discovered subgroups with the help of Vision-Language Models (VLLMs), and (3) leveraging Fisher information $\mathcal{I}_l$ to fine-tune the specified layers for bias mitigation.

incrementally retain singular values until the cumulative reconstruction error

$$\frac{\sum_{i=g+1}^{\min(S,M)} \sigma_i^2}{\|G_X\|_F^2} \leq \delta, \tag{1}$$

where $\delta$ is a predefined threshold for acceptable error. This adaptive rank selection allows us to identify the minimal set of subgroup bases that still sufficiently captures the variability of the data relative to the supervision signals.

By leveraging dynamic SVD in this manner, we effectively discover the inherent structure of multiple subgroups within the data, with each subgroup characterized by distinct basis vectors in $U$ that capture unique latent representations aligned with the black-box model's supervision signals. This approach enables a robust subgroup discovery process that is tailored to the complexity of the underlying data distribution.

Finally, we compute the cosine similarity between each sample's feature encoding and the subgroup basis vectors in $U$. The subgroup label for each sample is assigned by selecting the index corresponding to the maximum cosine similarity. By assessing model performance across the identified subgroups, we can uncover biased subgroups that exhibit lower accuracy.

### 3.3 Understanding the subgroup bias with VLLM

Inspired by Dunlap et al. (2024), we leverage Vision Large Language Models (VLLMs) to interpret subgroup differences and explain subgroup biases. The key challenge is handling the large number of images and subgroups, which complicates the reasoning of both within-group similarities and between-group differences. To address this, we introduce a progressive retrieval-based framework for bias interpretation.

For each subgroup, we maintain two core attributes: (1) the internal similarity within the subgroup, and (2) the distinguishing characteristics relative to other subgroups. We begin by randomly sampling $N$ images from different subgroups and prompting VLLMs to generate captions for each image. The VLLMs then

summarize the common features of images within the same subgroup, which are used to update a global similarity attribute.

Next, we concatenate images from the same subgroup into a single row and arrange images from different subgroups into separate rows, creating a composite image. Using the composite and the global similarity attribute, VLLMs identify the distinct characteristics between subgroups, updating a global difference attribute. After several iterations, we use the global difference attribute to provide an explanation for the subgroup biases.

### 3.4 Biased subgroup mitigation

We propose a novel approach that utilizes parameter importance scores to guide the bias mitigation process, aiming to reduce under-learning in biased subgroups while limiting over-learning in others.

Specifically, we compute the Fisher Information as an importance indicator for each layer $l$ in mitigating bias. This can be formulated as follows:

$$
\mathcal{I}_l = \frac{FI_l^b}{FI_l^{ub}},
$$
$$
\text{s.t.,} \quad FI_l^b = \mathbb{E}_{\theta^l}\left[ \mathbb{E}_{(x,y)\sim\mathcal{D}_b} \left[ \nabla_{\theta^l}^2 \mathcal{L}(f(x), y) \right] \right],
$$
$$
FI_l^{ub} = \mathbb{E}_{\theta^l}\left[ \mathbb{E}_{(x,y)\sim\mathcal{D}_{ub}} \left[ \nabla_{\theta^l}^2 \mathcal{L}(f(x), y) \right] \right],
\tag{2}
$$

where $\theta^l$ represents the parameters in layer $l$, $\mathcal{D}_b$ and $\mathcal{D}_{ub}$ denote the dataset containing identified biased and unbiased subgroups, respectively, $y$ is the superclass label of $x$, and $\mathcal{L}$ is the classification loss. A larger $\mathcal{I}_l$ indicates a higher sensitivity of the model's parameters to biased subgroups compared to unbiased subgroups.

Following previous work (Wang et al., 2020; Zhang et al., 2024a), we add an auxiliary subgroup classification head to the backbone of the biased neural network to help distinguish features among subgroups, aiding bias mitigation. Additionally, we apply LoRA to important layers where $\mathcal{I}_l > 1$. This step inherently increases the model's parameter count, thereby improving its ability to learn robust, generalized features, ultimately supporting the learning of under-represented subgroups. During the training process, we also scale the learning rate of different layers by their layer importance, balancing the representation learning between biased and unbiased subgroups.

## 4 Experiments

### 4.1 Setup

**Dataset**. Our experiments consist of four datasets: CIFAR-100 (Krizhevsky et al., 2009), Breeds (Santurkar et al., 2020), Hard ImageNet (Moayeri et al., 2022), and ICSD-171k (Salgado et al., 2023). Specifically, (1) the CIFAR-100 dataset consists of 20 superclasses, where each superclass can be divided further into 5 fine-classes. Each fine-class consists of 500 images for training and 100 for testing. (2) The Breeds dataset is a subset of the ImageNet-1K dataset. It is manually constructed and used to study the subgroup robustness, which has 13 superclasses with 10 fine-classes in each superclass. (3) The ICSD-171K dataset is an X-ray diffraction dataset. It consists of 7 crystals systems, which can be further unevenly divided into 230 space groups. (4) The Hard ImageNet dataset is another subset of the ImageNet-1K dataset, where the "hard" comes from the spurious correlation involved in the dataset, leaving the partition of the fine-class unknown.

**Baselines**. Our experimental framework encompasses three pivotal tasks: detecting multiple biased subgroups, interpreting these subgroups, and mitigating bias across them. In the detection phase, we benchmark our approach against SVM (Jain et al., 2022), Domino (Eyuboglu et al., 2021), and DIM (Zhang et al., 2024a). For understanding the bias, we leverage the GPT-4V to explain the discovered bias. For biased subgroup mitigation, we first annotate samples with pseudo labels generated by different detection methods. These labels are then leveraged by the supervised fine-tuning technique to improve the model's subgroup robustness.

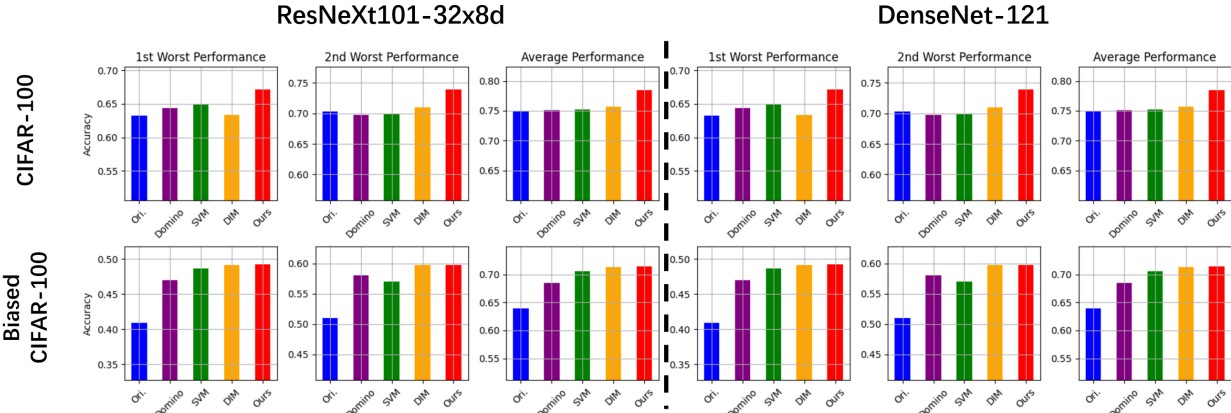

Figure 4: Mitigation results of two models on the CIFAR-100 dataset under two different settings. The original CIFAR-100 dataset consists of 20 superclasses, each containing an equal number of subclass images. The biased CIFAR-100 dataset is a transformed version, where two subclasses in each superclass are randomly selected, and their image counts are reduced to introduce a manual quantity bias, thereby influencing the learned model.

**Implementation details**. We study the ResNet and fine-tuned CLIP in our experiments, where we append a linear layer to the CLIP image encoder for classification. We train the two models on the CIFAR-100, Breeds, and Hard ImageNet datasets. We use 5 AlexNet as the surrogate model for bias discovery. The surrogate models are firstly respectively pre-trained on SVHN, Flower, Euro, DtD, and CIFAR-10 datasets, then fine-tuned on the targeted dataset, where the feature encoder is frozen during the fine-tuning and only the linear layer is updated.

### 4.2 Evaluation on biased subgroup discovery

We start with the experiments on biased subgroup detection. We apply the baseline methods and our method to discover and identify the biased subgroups, where the detection accuracy is reported in table 1. Compared with SVM, Domino, and DIM, our method can achieve a better performance in detecting the biased subgroups with a clear margin of 17.8%. On the in-distribution dataset of CLIP, *i.e.*, CIFAR-100 and Breeds, our method achieves an average accuracy of 82.5%. On the out-of-distribution

Table 1: The success rate of different methods on the biased subgroup detection task on three datasets, including CIFAR-100, Breeds, and ICSD-171K.

| Method | SVM | Domino | DIM | Ours |
|---|---|---|---|---|
| CIFAR-100 | 42.5 | 40.0 | 55.0 | **60.0** |
| Breeds | 42.3 | 40.4 | 61.5 | **65.4** |
| ICSD-171K | 17.4 | 19.1 | 20.0 | **23.5** |

dataset of CLIP, *i.e.*, the ICSD-171K, the feature encoder can not work well for accurate subgroup discovery at the clustering step of Domino and decomposition step of SVM and DIM, leading to a bad performance on biased subgroup detection. The inconsistent number of fine-classes in each superclass makes it challenging to accurately capture the subgroup indicator for baseline methods, which assume a fixed number of subgroups. By eliminating the reliance on CLIP with the use of multiple surrogate models on the studied dataset as well as the adaptive subgroup discovery, our method still achieves a good performance with 65.1% accuracy.

### 4.3 Evaluation on bias mitigation

We further conduct experiments on bias mitigation across three datasets: CIFAR-100, Breeds, and ICSD-171K. We evaluate the mitigation performance in two subtasks. First, following the approach of (Jain et al., 2022; Zhang et al., 2018), we assess classification accuracy across the worst-performing subgroups. Second, we evaluate the model's performance on the entire dataset. A good model should not only perform well on the

Table 2: The classification accuracy of ResNet-101-32x8d on the Breeds, and ICSD-171K test set. We present results on the worst four and three subgroups to study the mitigation performance of multiple biased subgroups. The overall accuracy is also provided for a comprehensive evaluation.

| Datasets | Breeds | | | | | ICSD-171K | | | |
|---|---|---|---|---|---|---|---|---|---|
| Method | 1 | 2 | 3 | 4 | Avg | 1 | 2 | 3 | Avg |
| Ori. | 56.3 | 65.5 | 70.2 | 73.9 | 77.1 | 5.7 | 13.6 | 17.3 | 76.5 |
| Domino | 58.1 | 67.4 | 74.5 | 75.3 | 79.4 | 8.4 | 16.7 | 18.9 | 76.9 |
| SVM | 57.9 | 67.4 | 73.2 | 74.6 | 79.3 | 8.1 | 15.2 | 19.5 | 76.7 |
| DIM | **59.5** | 68.1 | 75.6 | 76.3 | 81.2 | 10.6 | 19.0 | 21.8 | 79.2 |
| Ours | 59.3 | **70.9** | **76.5** | **77.8** | **81.1** | **13.9** | **21.6** | **23.6** | **82.1** |

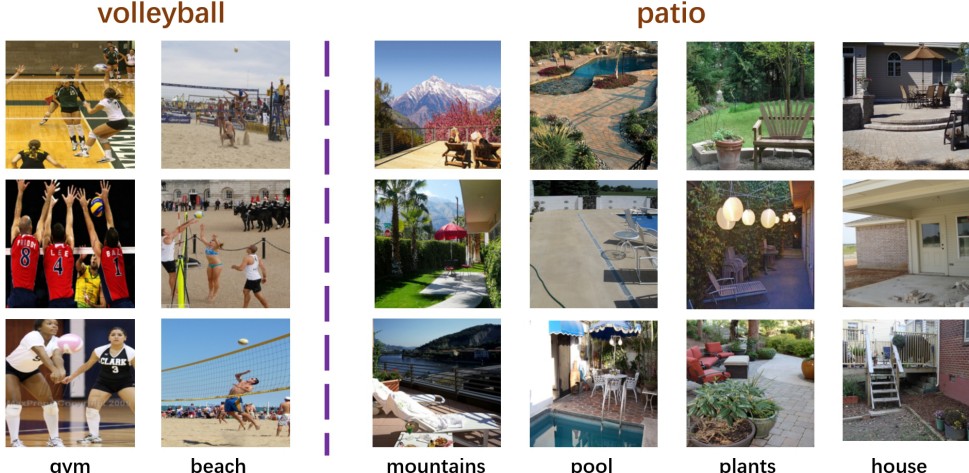

Figure 5: We use our method to study the subgroup bias in CLIP. By adaptive subgroup discovery, we find that there are two subgroups in "volleyball", and four subgroups in "patio".

worst subgroups but also demonstrate strong overall performance. The results are presented in fig. 4 and table 2.

It is important to note that each dataset presents unique challenges. The CIFAR-100 dataset, for instance, consists of only five subclasses per superclass. In addition to the standard setup, we follow the method in (Jain et al., 2022) to create a biased dataset by sampling from the CIFAR-100 dataset, where two subclasses within each superclass have only 20% of the samples compared to the other subclasses. In contrast, the Breeds dataset contains more subclasses, which challenges the scalability of baseline methods. The ICSD-171K dataset, being out-of-domain with respect to natural images, is particularly difficult for the CLIP model to recognize. Furthermore, the number of subclasses varies across superclasses, which poses additional challenges in terms of both generalization and scalability.

Our method achieves the best performance in bias mitigation across these diverse datasets. Specifically, compared to SVM, Domino, and DIM, our method provides more accurate pseudo-subgroup labels for supervised mode-centric mitigation methods. This results in average improvements in worst subgroup accuracy of up to 0.83% on CIFAR-100, 0.75% on the Breeds dataset, and 2.4% on ICSD-171K.

### 4.4   Studying the subgroup bias on Hard ImageNet learned by CLIP

We use our method to discover the subgroup bias in CLIP. We select two signals for the supervision of decomposition, 1) we use logits of the fine-tuned classification head which is additionally appended to the

image encoder; 2) we use the cosine similarity between the image features and text features of paired class labels. Results are depicted in fig. 5.

We can see that there are different numbers of subgroups in different classes on the Hard ImageNet dataset detected by the CLIP. By summarizing the key difference between different subgroups using GPT-4 Vision (Achiam et al., 2023), it can be seen that the "volleyball" class can be further divided into two subgroups according to its sports scenes, including the "gym" and "beach". Also, the "patio" class can be divided into four subgroups based on its surrounding objects, namely the "mountain", "pool", "plants", and "house". The discovery of subgroups in CLIP by studying the Hard ImageNet dataset suggests that CLIP is also influenced by multiple subgroups, even with the weak supervision of description during the contrastive learning process.

### 4.5 Abaltion study

**On the use of** $\delta$. In our work, we leverage $\delta$ in eq. (1) to control the decomposition granularity of subgroup discovery. Here, we conduct additional experiments on the CIFAR-100 and Breeds datasets to study the impact of $\delta$ on the performance of discovering biased subgroups. The results are reported in table 4.

In general, a larger value of $\delta$ results in fewer subgroups being discovered, which could lead to the oversight of biased subgroups. Conversely, a smaller $\delta$ leads to the discovery of more subgroups, enabling the identification and clustering of finer-grained features, thus increasing the potential to uncover more biased subgroups. The reported results validate this trend. Specifically, setting $\delta = 5\%$ strikes a balance, achieving good performance by uncovering most biased subgroups while maintaining low computational costs.

Table 3: Studying the impact of $\delta$ on the performance of mining biased subgroups.

| $\delta$ | 1% | 5% | 10% | 15% | 20% |
|---|---|---|---|---|---|
| CIFAR-100 | 60.0 | 60.0 | 52.5 | 50.0 | 45.0 |
| Breeds | 67.3 | 65.4 | 65.4 | 57.5 | 55.7 |

**On the use of** $\mathcal{I}_l$. While previous works fine-tune the entire neural network for bias mitigation, we propose using Fisher information, as shown in eq. (2), to first identify the important layers and then fine-tune only those layers for bias mitigation, thus avoiding further over-learning of unbiased subgroups. To demonstrate the benefits of using Fisher information, we conduct experiments where this technique is disabled, relying solely on the adaptive subgroup discovery method to generate pseudo-labels for normal bias mitigation fine-tuning. We report the changes in accuracy for both the worst and best-performing subgroups across different datasets to illustrate the impact of this technique. The results show that when specialized layer fine-tuning with Fisher information is disabled, the model tends to perform worse on the worst-performing subgroups and continues to improve on the well-performing subgroups, resulting in a larger gap between the subgroups.

Table 4: Studying the impact of $\mathcal{I}_l$ on the performance of bias mitigation. We report the changes of the accuracy compared with the method using $\mathcal{I}_l$.

| Datasets | CIFAR-100 | Breeds | ICSD-171K |
|---|---|---|---|
| Worst | +0.1 | −0.2 | −0.1 |
| Best | +1.3 | +0.9 | +3.7 |
| Avg | +0.5 | +0.4 | +1.1 |

## 5 Conclusion

In this paper, we study an important problem, the discovery and mitigation of multiple biased subgroups. Compared with previous studies, we make an effort to solve two challenging tasks, including the discovery of inconsistent numbers of subgroups across different classes and the mitigation of novel subclasses that are not present in the local dataset but exist in the wild. Also, previous studies are limited to the use of CLIP embedding for bias discovery, while neglecting the fact that CLIP is also biased towards the spurious correlation, leading to an unreliable explanation in the whole pipeline. To address these problems, we propose to use the ensemble of surrogate models fine-tuned on the targeted dataset to replace the CLIP for feature encoding while minimizing the impact of bias. The encoded features are used for subgroup discovery and

pseudo-subgroup label generation, as well as bias mitigation. Extensive experiments on CIFAR-100, Breeds, and ICSD-171K dataset sufficiently demonstrate the effectiveness of our proposed method.

**Limitations**. While different methods can be used for subgroup discovery, there still is a lack of trustworthy metrics for performance evaluation due to the existence of multiple kinds of biases. Also, there remains a problem with how to discover multiple biased subgroups in multi-modality data and models.

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
