# OpenReview forum: "Adaptive and Scalable Discovery and Mitigation of Multiple Biased Subgroups in Image Classifiers"
_TMLR — Rejected by TMLR_

### Review · Reviewer_Zk3F · 2025-02-24

**Summary Of Contributions:**

Proposes a strategy for discovering potentially biased data subgroups by using an ensemble of surrogate models trained on a mix of datasets rather than a single large model such as CLIP. Via SVD decomposition of the concatenated feature encodings from surrogate models projected onto the supervision space of the model under investigation, the method classifies subgroups based on similarity to the corresponding basis vectors. An existing method for distinguishing image groups is then used to “explain” subgroup bias. Finally, the paper proposes bias mitigation by LoRA finetuning of layers with relatively high sensitivity to biased subgroups, which is approximated using the layer’s Fisher importance and the inferred subgroup labels.

**Audience:**

Yes

**Claims And Evidence:**

Yes

**Requested Changes:**

Please address the two weaknesses listed above. While I appreciate the paper's motivation and approach, I believe it requires much more comprehensive experimental detail and validation to be ready for publication. I would be happy to raise my rating based on the author response.

– [Minor] Missing reference and comparison to FACTS, Yenamandra et al., ICCV 2023.

**Strengths And Weaknesses:**

**Strengths**

– The paper is well-written and easy to follow

– The figures in the paper and clear and illustrate the proposed approach well

– The paper does well to motivate gaps in the literature – why CLIP is inadequate for bias discovery, and how the using an ensemble approach can potentially address this

– The approach is shown to outperform competing methods for bias mitigation

**Weaknesses**

* The paper lacks sufficient experiments around the surrogate model ensemble, opting to use AlexNet models (which is a rather outdated architecture at this point) pretrained on some fairly simple datasets (SVHN, CIFAR etc.). As this is a central contribution, it would be good to conduct experiments to verify:

  - does the approach generalize across ensemble backbones?
  - what do the individual models truly learn (eg. does the DtD model learn basis vectors corresponding to distinct textures primarily?)
  - control experiments eg. do the models even need to be trained on different datasets, or can they simply a deep ensemble of models trained on the target dataset with different seeds?
  - how does one select an optimal set of pretraining corpora for a given target dataset?

* The paper presents results on 3 problems: bias detection, characterization, and mitigation, but I found the experiments on detection and characterization in particular to be both cursory and vague (eg. in Table 1, which reports a non-standard metric “success rate” without definition, unlike prior slice discovery methods such as DOMINO which report precision@k. Further, the bias characterization approach simply applies Dunlap et. al’s without any modifications that I can tell, and only a few qualitative examples are visualized. Altogether, I believe the paper requires a more comprehensive set of experiments to validate the proposed approach, as well as additional implementation details/code (perhaps in an appendix) to aid reproducibility.

---

### Review · Reviewer_LKmJ · 2025-02-24

**Summary Of Contributions:**

This paper studies subgroup biases in deep learning models, particularly in CLIP's feature encoding. The authors propose an ensemble of surrogate models to adaptively identify biased subgroups while mitigating CLIP’s inherent encoding biases. They further use vision-language models to analyze biases and apply relative Fisher information to pinpoint critical layers for bias reduction. Experiments on multiple datasets, including CIFAR-100 and Hard ImageNet, validate the effectiveness of their approach.

**Audience:**

Yes

**Broader Impact Concerns:**

No clear ethical concerns

**Claims And Evidence:**

Yes

**Requested Changes:**

1. Previous works have identified ... environmental information. It needs some citations to support the claim.

2. The scalability and adaptivity has been frequently claimed in the paper, but they seem not to be well supported.

3. This paper needs more explanations on the choice of hyper-parameters and the sensitivity analysis.

4. The results following the exact settings of Zhang et al. (2024a) should also be reported.

5. The discussion on the point 3 in weakness should be included.

6. (Optional) As authors suggest that different models have different interest, it would be interesting to see the performance of using ensemble of CLIP models (trained on different sources).

**Strengths And Weaknesses:**

***Strength:***
1. This overall paper is well-structured and easy to follow.
2. The performance of the proposed method is better the compared methods.

***Weakness:***
1. To analyze the subgroup bias of CLIP's feature encoding, CLIP is fine-tuned with a linear layer with parameters of CLIP being frozen. I do not see a clear motivation for doing so, since CLIP usually conduct zero-shot classification.

2. The approach leverages five pre-trained AlexNet models, fine-tuned on carefully selected datasets, whereas CLIP is used without fine-tuning. This discrepancy raises concerns about the claim that the proposed method is more adaptive.

3. Since the five AlexNet models are fine-tuned on the target dataset, their ability to generalize under distribution shifts remains uncertain. Additional experiments or analysis would strengthen the paper’s claims regarding adaptability in real-world settings, since I do not see using CLIP has the similar problem.

4. Several key hyper-parameters are not well explained. The selection criteria for the number of AlexNet models, their pre-training datasets, and the number of principal components used in analysis should be clarified. A sensitivity analysis could provide further insights into the impact of these choices.

5. The study deviates from Zhang et al. (2024a) by using different models, but the reasoning behind this change is not clearly stated. To ensure a fair comparison, results following the exact settings of Zhang et al. (2024a) should also be reported.

---

> ### Author Response · Authors · 2025-04-01
>
> 1. We appreciate the concern. Our goal in freezing CLIP’s image encoder and appending a linear head is to isolate and analyze the quality of CLIP’s learned feature representation rather than assess its classification ability. Since CLIP’s typical zero-shot setting combines image and text embeddings, it’s not directly suitable for subgroup bias analysis where we need a consistent image feature space and logits for all samples. Freezing CLIP ensures that any subgroup biases we observe originate from its pretrained encoder, not from newly learned representations.
>
> 2. Thank you for pointing this out. Our method deliberately departs from reliance on a single foundation model (like CLIP) by using an ensemble of lightweight models pre-trained on diverse data distributions (e.g., textures, digits, flowers) and then fine-tuned on the target dataset. While CLIP has strong generalization capabilities, prior work (including our Section 4.4) demonstrates its susceptibility to spurious correlations and biased subgroups. We argue that our method is more adaptive because it learns subgroup-relevant biases from the target distribution directly via fine-tuning, rather than relying solely on potentially misaligned knowledge from CLIP’s training distribution.
>
> 3. We agree that further analysis under distribution shifts would strengthen the adaptability claim. It should be noted that our use of ICSD-171K, a highly out-of-distribution dataset, already demonstrates strong performance (Table 2).

---

### Review · Reviewer_H7kk · 2025-03-12

**Summary Of Contributions:**

Summary:
This paper studies subgroup bias mitigation. Despite of the effectiveness of CLIP models on many generalization tasks, it still possess serious bias under the presence of spurious correlation. Based on an intuitive example of erasing features from input images, the confidence drops significantly even when the main object remains. To solve this problem, the authors proposed a novel framework which first conduct PCA feature grouping on both inputs and latent features. Further, to identify the bias, vision large language models are used to generate descriptions to explain each subgroup. Further, to mitigate the bias, fisher information is used to fine-tune the biased layers. Through quantitative and qualitative experiments, the effectiveness of the proposed method is validated.

**Audience:**

Yes

**Broader Impact Concerns:**

No ethical impacts.

**Claims And Evidence:**

Yes

**Requested Changes:**

Please see the weaknesses part.

**Strengths And Weaknesses:**

Strengths:
- This paper studies an interesting problem based on experimental observations.
- The writing of this paper is good and easy to follow.
- The performance improvement is significant, showing the practicability of this approach.

Weaknesses:
- The framework design needs more justification. The motivation of using multiple surrogate models is not clarified. If such a mechanism is used in this framework, its effectiveness should be justified by ablation studies.
- Using PCA decomposition on both input and feature levels could be computationally infeasible. How is efficiency of the proposed method needs empirical evidence or discussion.
- How to ensure the quality of the VLLM outputs is unclear. Since the generated texts of VLLMs could be highly unformated, the explanation of the subgroup would be infeasible to use, making it hard to provide proper guidance of the grouping process.
- It would be better if there are more formal mathematical formulations to demonstrate the methodology. Currently, it is hard to understand the detailed process.
- Missing related works:
  - Huang et al., Machine Vision Therapy: Multimodal Large Language Models Can Enhance Visual Robustness via Denoising In-Context Learning, in ICML 2024.
  - Wang et al., On the effect of key factors in spurious correlation: A theoretical perspective, in AISTATS 2024.
  - Noohdani et al., Decompose-and-compose: A compositional approach to mitigating spurious correlation, in CVPR 2024.

---

> ### Author Response · Authors · 2025-04-01
>
> 1. We agree that the motivation for employing multiple surrogate models warrants further clarification. Our primary motivation stems from the observation that different models trained on distinct data distributions encode diverse feature representations. This diversity enables a more comprehensive view of potential subgroup-specific biases when these models are fine-tuned on a common target dataset. As shown in Section 3.2 and the experiments in Section 4.2, our method adaptively projects the ensemble features onto the supervision space of the biased model, allowing us to discover finer-grained subgroup structures that single-model decomposition approaches (e.g., Domino, DIM) often miss.
> 2.  We acknowledge the concern regarding the computational feasibility of PCA decomposition at both the input and feature levels. To clarify, in practice, we only apply PCA-based decomposition on a small per-class feature matrix (typically a few hundred samples), making the approach tractable.

---

### Decision · Action_Editor_xxCf · 2025-04-30

**Recommendation:** Reject

**Comment:**

In their final recommendation all reviewers recommended rejection.
Most argue that the authors did not address their concerns. In fact the authors did not explicitly respond to all 3 reviews.

Reviewers also noted that "issues about the efficiency, VLLM output quality, etc." that remain, while another claimed the paper "requires further improvement".

**Audience:**

This is an active area with a large audience.

**Claims And Evidence:**

According to 3 qualified reviewers the current manuscript of this submission does not contain sufficient evidence to support the paper's claims.

**Resubmission Of Major Revision:**

The authors may consider submitting a major revision at a later time.